# MEMORY-EFFICIENT REINFORCEMENT LEARNING WITH PRIORITY BASED ON SURPRISE AND ON-POLICYNESS

## ABSTRACT

In off-policy reinforcement learning, an agent collects transition data (a.k.a. experience tuples) from the environment and stores them in a replay buffer for the incoming parameter updates. Storing those tuples consumes a large amount of memory when the environment observations are given as images. Large memory consumption is especially problematic when reinforcement learning methods are applied in scenarios where the computational resources are limited. In this paper, we introduce a method to prune relatively unimportant experience tuples by a simple metric that estimates the importance of experiences and saves the overall memory consumption by the buffer. To measure the importance of experiences, we use *surprise* and *on-policyness*. Surprise is quantified by the information gain the model can obtain from the experiences and on-policyness ensures that they are relevant to the current policy. In our experiments, we empirically show that our method can significantly reduce the memory consumption by the replay buffer without decreasing the performance in vision-based environments.

## 1 INTRODUCTION

Reinforcement learning (RL) has become a promising approach for learning complex and intelligent behavior from visual inputs (Mnih et al., 2016; Kalashnikov et al., 2018). In particular, off-policy RL algorithms (Mnih et al., 2015; Hessel et al., 2018) generally achieve better sample efficiency than on-policy algorithms by using experience replay (Lin, 1992). In experience replay, the transitions observed in the environment are stored as *experience tuples* in a replay buffer and used repeatedly. In addition, the replay buffer has the role to remove the correlations between the samples in a mini-batch. However, these methods require a significant number of experience tuples, which consume a large amount of memory when the observations are given as images.

Many prior studies on replay buffers in RL consider how the experience tuples are sampled from the buffer (Schaul et al., 2016; Zha et al., 2019; Fujimoto et al., 2020; Sun et al., 2020; Oh et al., 2021). If we are to train an agent in a scenario where the available resources are limited, the replay buffer needs to be reduced to an appropriate size. It is known that simply reducing the size of the replay buffer will lead to unexpected performance degradation (Liu & Zou, 2018; Fedus et al., 2020). There is some prior work on how to select old experience tuples to overwrite when a new experience tuple comes into a relatively small buffer (Pieters & Wiering, 2016; de Bruin et al., 2016b; 2018). However, they do not consider a memory-efficient method for image observation where memory consumption is large. We aim to reduce the size of the replay buffer without degrading the performance in visual domains.

Our intuition is that some experience tuples are important for gaining knowledge about the environment and others are not. For example, the scenes in a video game that do not accept any inputs from the player, such as the standby screen, occupy a considerable amount of time in the game, but they do not provide much information. In contrast, the frames that are within a few frames of the scenes where the player earns or loses points are often important in the game. In particular, the scenes that are related to the end of a gameplay are important to keep the game going and to obtain high scores.

On the basis of this intuition, we propose to prioritize and keep experience tuples that are deemed important and discard the others. The overview of our approach is shown in Figure 1. In this paper,

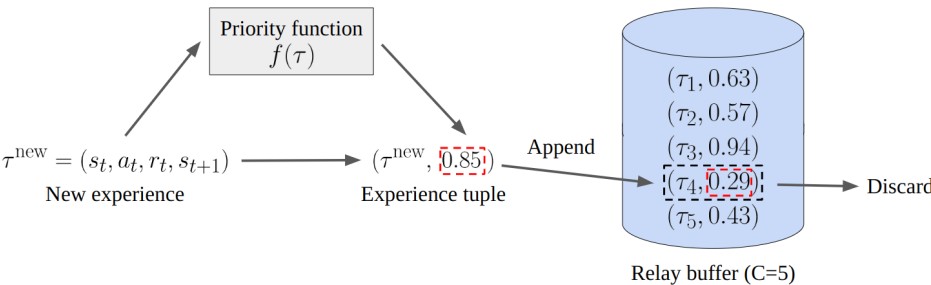

Figure 1: Illustration of our proposed method. Priority of the experience tuples are calculated, and they are discarded based on the priority when a new experience tuple arrives. The capacity of the replay buffer in the figure is small for the sake of readability.

we propose a metric based on *surprise* (Itti & Baldi, 2005) and *on-policyness* (Fedus et al., 2020) to estimate the importance of the experiences and to prune the unnecessary experience tuples stored in the replay buffer. We hypothesize that the importance of an experience tuple is determined by the degree of information that the model gains by obtaining the experience tuple and the strength of relevance to the current policy. Surprise is related to the uncertainty of an experience tuple and represents the novelty of the information. At the same time, however, the agent does not need to keep all the experience tuples that have high uncertainty, especially the ones that are likely to be outliers. The on-policyness metric is introduced to suppress the adverse effects of those outliers and keep the experience tuples close to the actual transition the current agent takes. We demonstrate that our method can be implemented as simple modifications to the existing implementations for replay buffers. In addition, our approach can be combined with the existing approaches that are used to sample experience tuples from the buffer.

We show in the experiments that the proposed method of data pruning can save the memory consumption of the buffer and prevent a performance decrease when the size of the buffer is limited.

## 2 RELATED WORK

Experience replay using replay buffers is a common method in off-policy RL algorithms to make the most of the experiences obtained by the behavior policy (Mnih et al., 2015; Haarnoja et al., 2018; Hessel et al., 2018). Most of the research on replay buffers focuses on how the experience tuples are sampled from the buffer when creating a mini-batch to accelerate the training process. The most well-known is the prioritized experience replay (PER) (Schaul et al., 2016), which prioritizes the sampling of the experience tuples based on their training errors. There have been many studies investigating other effective indicators for sampling the experience tuples to achieve efficient training. Some used fixed metrics to select experience tuples (Schaul et al., 2016; Fujimoto et al., 2020; Sinha et al., 2022), while others constructed and trained models to select the tuples which maximize the improvement of the policy model after the parameter update (Zha et al., 2019; Oh et al., 2021). There have also been some studies examining experience selection methods based on indicators such as reward (Pieters & Wiering, 2016), surprise (de Bruin et al., 2018), and exploration (de Bruin et al., 2016a; 2018), and how they affect the performance of an RL agent in state-based environments using a relatively small buffer. Chen et al. (2021) proposed a method that aims to construct a memory-efficient RL algorithm in vision-based domains. In this method, they freeze the parameters in the convolutional neural network (CNN) encoder at the early stage of the training and store the latent vector from the encoder, instead of storing raw images to save memory consumption. The major difference in our approach is that we choose the experiences to discard. Since the discarding of the experiences is done independently of the sampling process, the sampling methods used in the replay buffer can be combined with our method.

We introduce a method to select experience tuples in RL settings. Determining which tuples to keep can be seen in a continual learning setting (Lopez-Paz & Ranzato, 2017; Shin et al., 2017; Isele & Cosgun, 2018; Rolnick et al., 2019). In these settings, the model is given a stream of data and the training is done in an online manner. These settings suffer from catastrophic forget-

ting (French, 1999; Goodfellow et al., 2013), where the model forgets the information observed in the early stage of the training. One method to prevent this issue is to keep some of the data in a buffer to remember the key components in each domain. In this approach, metrics like the gradient of the samples (Lopez-Paz & Ranzato, 2017; Aljundi et al., 2019), the feature vectors (Rebuffi et al., 2017), or the learnability of the data (Sun et al., 2022) are used to select the data to keep. Tasks from different domains are given as the training proceeds in continual learning in RL (Isele & Cosgun, 2018; Rolnick et al., 2019). The agents need to remember the previous task after being introduced to new tasks or have to adapt quickly using the knowledge from the previous tasks. In our setting, we only consider a single task domain and do not require adaptation to a different task. Catastrophic forgetting is thus not as important as in these methods in our setting.

## 3 BACKGROUND

We consider a Markov decision process (MDP), defined by $(\mathcal{S}, \mathcal{A}, \mathcal{T}, \rho_0, r, \gamma)$, where $\mathcal{S}$ is a state space, $\mathcal{A}$ is a set of actions, $\mathcal{T}$ is a state transition function, $\rho_0$ is a distribution over the initial states, $r : \mathcal{S} \times \mathcal{A} \to \mathbb{R}$ is a reward function, and $\gamma \in [0, 1)$ is a discount factor. At each time step $t$, the agent observes the current state $s_t$ of the environment, selects and takes an action $a_t$ based on its policy $\pi(a_t|s_t)$. The environment returns an immediate reward $r_t$ and the next state $s_{t+1}$. The goal of the agent is to learn a stationary policy that maximizes the expected discounted sum of reward $R_t = \mathbb{E}^\pi[\sum_{t=0}^\infty \gamma^t r(s_t, a_t)]$, where the expectation is calculated over the trajectories sampled from $s_0 \sim \rho_0, a_t \sim \pi(\cdot|s_t)$, and $s_{t+1} \sim \mathcal{T}(s_t, a_t)$ for $t \geq 0$. $\tau_t$ is an experience tuple $(s_t, a_t, r_t, s_{t+1})$ which contains elements observed within a single time step in the environment.

### 3.1 DEEP REINFORCEMENT LEARNING

The mainstream of off-policy RL algorithms derives from $Q$-learning (Watkins & Dayan, 1992), which learns an action-value function called a $Q$-function. The $Q$-function is defined as a discounted sum of rewards, after taking an action $a$ on a state $s$:

$$Q^\pi(s, a) = \mathbb{E}^\pi \left[ \sum_{t=0}^\infty \gamma^t r(s_t, a_t) | s_0 = s, a_0 = a \right].$$

The $Q$-function is learned by minimizing the temporal difference (TD) error $\delta(\tau)$ defined as:

$$\delta(\tau) = (Q(s_t, a_t) - Q^{target}(s_t, a_t))^2,$$
$$Q^{target}(s_t, a_t) = r(s_t, a_t) + \gamma \max_a Q(s_{t+1}, a).$$

The deep $Q$-Network (DQN) algorithm (Mnih et al., 2015) approximates a $Q$-function with a deep neural network. The $Q$-function has been expressed and learned in various forms, for example, as a combination of a value function and an advantage function (Wang et al., 2016), or as a function returning the distribution of the $Q$-values (Bellemare et al., 2017; Dabney et al., 2018). In the distributed settings, $Q$-values are represented as a discrete probability distribution $p(Q)$, and commonly the TD error is calculated as the KL-divergence between the current and the target $Q$-value distributions (Bellemare et al., 2017):

$$\delta(\tau) = D_{\text{KL}}(p(Q^{target}(s_t, a_t))||p(Q(s_t, a_t))).$$

### 3.2 EXPERIENCE REPLAY

In an online setting, the collected experience tuples are discarded once they are used for updating the parameters. However, it is difficult for the model to learn from a single update loop, and thus discarding the experience tuples that can be rarely encountered can lead to insufficient training. Experience replay (Lin, 1992) is a method that uses a replay buffer to store experience tuples. A replay buffer is usually implemented with a ring buffer storing a fixed number of the latest experience tuples collected by the behavior policy. When the buffer is full, the oldest experience tuple is discarded and the new tuple is appended. By using the experience tuples repeatedly, experience replay improves the sample efficiency of the training algorithm. The simplest and the most common way of creating a batch is a uniform sampling from the buffer. The replay buffer also plays a role in suppressing the temporal correlation of online samples and preventing catastrophic forgetting.

---

**Algorithm 1** Proposed method with $Q$-learning

---

1: Initialize action-value network $Q$ with parameters $\theta$
2: Initialize replay buffer $\mathcal{D}$ with capacity $C$
3: **for** each time step $t$ **do**
4:     Select and execute action $a_t \leftarrow \arg\max_a Q_\theta(s_t, a)$
5:     Receive reward $r_t$ and next state $s_{t+1}$
6:     Calculate the pruning priority $f(\tau_t)$ of the experience tuple $\tau_t$
7:     **if** $t \geq C$ **then**
8:         $(\tau^{min}, f(\tau^{min})) \leftarrow \arg\min_{(\tau, f(\tau)) \in \mathcal{D}} f(\tau)$
9:         **if** $f(\tau^{min}) < f(\tau_t)$ **then**
10:             $\mathcal{D} \leftarrow \mathcal{D} \backslash \{(\tau^{min}, f(\tau^{min}))\}$
11:             Store the experience tuple $(\tau_t, f(\tau_t))$ into the replay buffer $\mathcal{D}$
12:         **end if**
13:     **else**
14:         Store the experience tuple $(\tau_t, f(\tau_t))$ into the replay buffer $\mathcal{D}$
15:     **end if**
16:     Sample a batch $\mathcal{B}$ from $\mathcal{D}$
17:     Update $\theta$ with $\mathcal{B}$
18:     Update the priority value in $\mathcal{B}$ and return it to $\mathcal{D}$
19: **end for**

---

## 4 METHOD

In this section, we present our method for prioritizing the experience tuples by their importance and determining the tuples to be retained according to the calculated priority values. This is used to ensure that the number of tuples fits the capacity of the limited-sized replay buffer. Our method will save the memory required and important experience tuples will be sampled much more frequently, which can result in acceleration of the training. We define the priority calculation function $f(\tau)$ for experience tuple $\tau_t$ as follows:

$$f(\tau_t) = w(s_t, a_t | Q) \delta(\tau_t),$$

where $\delta(\tau)$ is a value based on *surprise* and $w(s_t, a_t | Q)$ is a weight based on the current agents policy calculated using the $Q$-function. The details of the value and the weight will be described in the following sections.

### 4.1 SURPRISE

Surprise is a measure of how unexpected the experience tuples are for the model. Surprised-based scores have been used for exploration in reinforcement learning to efficiently find states that are not fully known to the model (Schaul et al., 2016; Burda et al., 2019; Berseth et al., 2021). Prioritizing experience tuples that have high uncertainty and sampling on the weighted distribution have been shown to accelerate training in deep RL (Schaul et al., 2016; Hessel et al., 2018). TD-errors are usually used as the uncertainty measure. The experience tuples with larger errors are more unexpected for the model and can have more room left to learn. In distributed $Q$-learning, TD-errors can be viewed as an approximation of the information gain that can be obtained from the experience tuple $\tau$:

$$
\begin{aligned}
I(Q, \tau) &= \mathbb{E}_\tau \left[ D_{\text{KL}}(p(Q|\tau) || p(Q)) \right] \\
&\simeq \mathbb{E}_\tau \left[ D_{\text{KL}}(p(Q^{target}) || p(Q)) \right] \\
&= \mathbb{E}_\tau [\delta(\tau)].
\end{aligned}
$$

In the second line, we used the assumption that the distribution of the original $Q$-value given the experience tuple will become close to the target $Q$-value distribution. This is because the $Q$-function is trained to minimize the difference between the target $Q$-function using the experience tuple $\tau$. Selecting tuples with a large TD-error will result in choosing tuples that have large information gain.

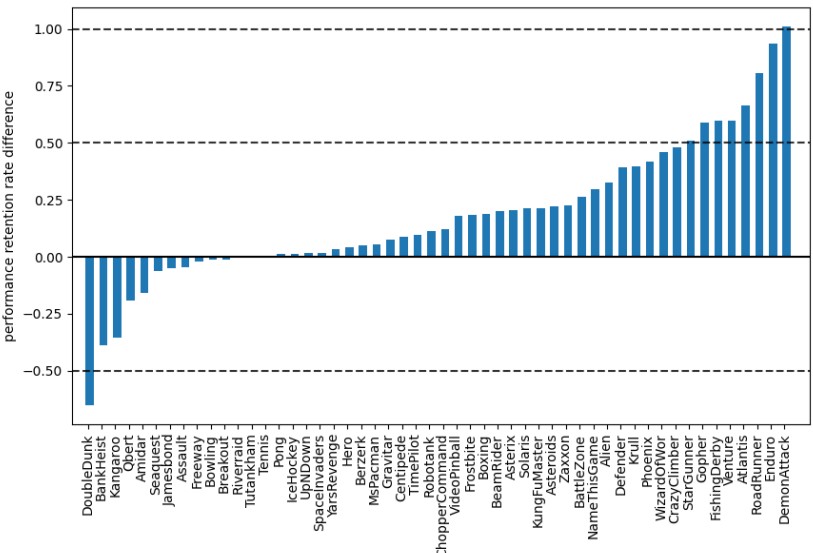

Figure 2: The best evaluation performance difference between the proposed method and the baseline method in settings where the replay buffer size is constrained to 10k. The results are normalized by the performance of the unconstrained baseline method.

## 4.2 ON-POLICYNESS WEIGHT

Selecting experience tuples only from those with large errors will result in unstable training. This is because the tuples that are far from the state and action distribution of the current policy tend to have a large error since they are not seen frequently. To overcome this issue, we additionally introduce on-policyness (Fedus et al., 2020) to the priority metric. Here, on-policyness is defined as how much the experience tuple stored in the buffer reflects the current target policy.

The term of on-policyness is important in on-policy RL algorithms where the behavior policy and the target policy need to be the same. A common way to satisfy this condition is to constrain the target policy to stay close to the behavior policy (Schulman et al., 2015; 2017). The on-policyness of an experience tuple is also important in off-policy training (Zhang & Sutton, 2017; Hausknecht & Stone, 2016; Novati & Koumoutsakos, 2019). To reflect the on-policyness of the experience tuples to the priority values, we propose a weight function in addition to the TD-error:

$$w(s_t, a_t|Q) = \frac{\exp(Q(s_t, a_t))}{\sum_a \exp(Q(s_t, a))}.$$

This weight function $w(s, a|Q)$ reflects how likely the model is to take an action $a_t$ at a certain state $s_t$ and can be viewed as a soft $Q$-policy (Haarnoja et al., 2017). The weight is always set to 1 when the priority is calculated for the first time to ensure that every transition will be used at least once.

At first, we considered using a hard policy for measuring the on-policyness. The weight function with a hard policy returns 1 if the action $a$ taken in the experience tuple matches the action that maximizes the $Q$-value on the state $s$ in the experience tuple, and returns a small value $\epsilon$ for tuples containing other actions. This approach is similar to the action probability distribution of an epsilon-greedy policy. However, using soft weights performed slightly better in the initial experiments, so we did not use the hard policy for evaluation.

## 4.3 OVERALL ALGORITHM

The overall algorithm is shown in Algorithm 1. In our algorithm, the priority of each experience tuple is calculated with the function explained above. A priority value $f(\tau_t)$ for pruning experience tuples is calculated after the experience $\tau_t$ is sampled from the environments. We add the priority value to the tuple $(\tau_t, f(\tau_t))$ stored in the buffer. The priority values of tuples are updated only when they are sampled to create a minibatch.

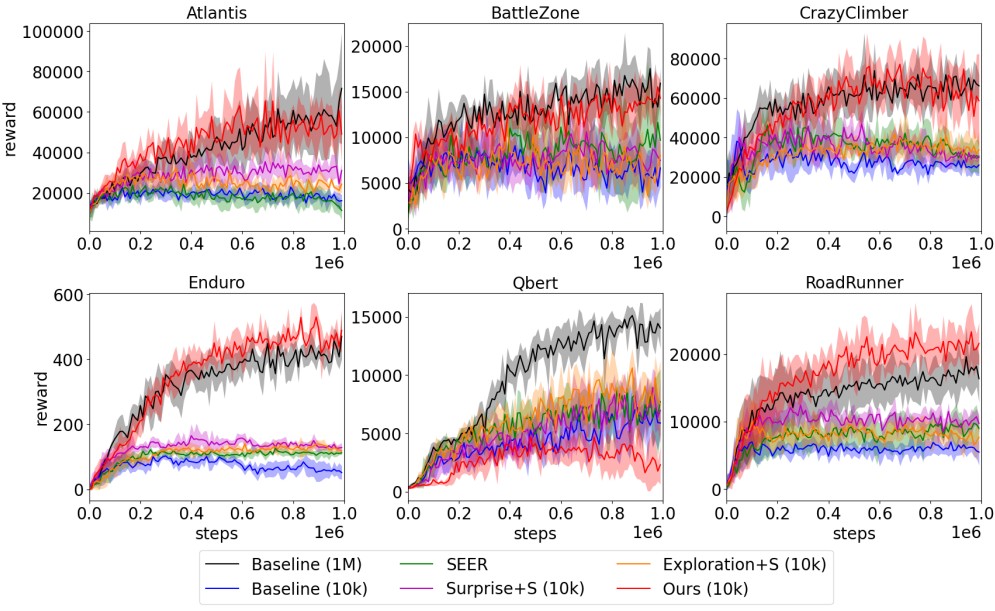

Figure 3: Comparison of the cumulative rewards in constrained-memory settings. The solid line and shaded regions represent the mean and standard deviation, respectively, across five random seeds.

The process of experience selection is executed when the buffer is full and a new experience tuple arrives. The tuple with the minimum priority value is discarded. These priority values are used and calculated independently of the ones used in PER. We also considered selecting tuples stochastically instead of selecting them deterministically but found it difficult to adjust the hyperparameters to provide appropriate weighting to select the tuples.

## 5 Experimental results

### 5.1 Experimental settings

We conducted experiments on 52 Atari (Bellemare et al., 2013) games to investigate the efficacy of our method. Atari is a set of commonly used benchmark environments for visual discrete control. It contains 57 games in total. In our experiments, we used 52 out of 57 environments. We excluded five games from the evaluation because they only provide sparse rewards or require a significant amount of exploration. Those environments are challenging even for the baseline algorithm without any constraint, and since our method does not promote exploration before the baseline algorithm receives an initial reward signal from the environments, we considered them not appropriate for our comparison purpose.

We trained our agent using the Rainbow (Hessel et al., 2018) algorithm for 1M training steps with the buffer size constrained to 10k. We ran experiments and compared our methods with the following experience selecting approaches. **Baseline**: discards the experience tuples in the First In First Out (FIFO) manner. **Surprise+S** (de Bruin et al., 2018): chooses the experience tuples stochastically using the same sampling method as PER. The sampling priority is based on the inverse of the TD-error $\delta(\tau)$. This sampling makes tuples with a small TD-error more likely to be discarded. **Exploration+S** (de Bruin et al., 2018): chooses the experience tuples with rare actions. The original implementation used action probabilities which cannot be directly used for Rainbow training. Instead, we used the sampling priority based on soft-policy metrics and sampled the tuples stochastically in the same way as PER. This sampling makes the tuples with rare actions stay longer in the buffer. In addition to these experience selection methods, we added results using **SEER** (Chen et al., 2021), which is a data compression method. We also experimented with the setting where no constraints are placed on the buffer size in the baseline method to show the upper bound of the performance. The details of the model architecture and hyperparameters are shown in Appendix A.

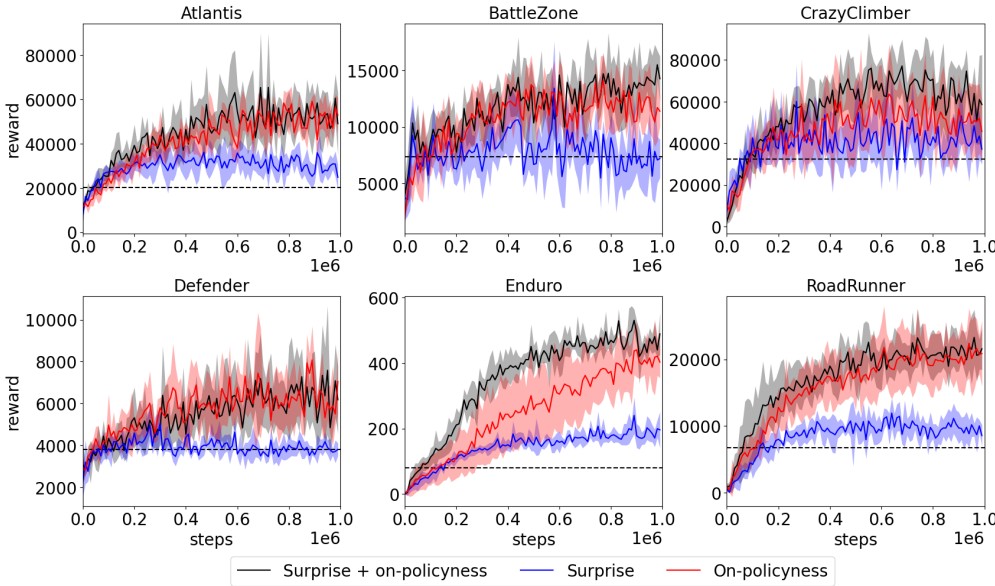

Figure 4: Comparison of the cumulative rewards when each factor was removed from the priority calculation function in constrained-memory settings. The solid line and shaded regions represent the mean and standard deviation, respectively, across five random seeds. The dotted line shows the best mean score of the baseline method in the constrained setting.

We run experiments on five random seeds for each environment. The performance of the agent on each seed is evaluated by taking the mean cumulative reward of five evaluation episodes every 10k training steps.

## 5.2 RESULTS

We summarize the performance in each Atari environment in Figure 2. We found that our experience selection method suppresses the degradation of the agent performance when the size of the replay buffer is limited. In 39 out of the 52 environments, the proposed method has shown improvement from the baseline results. Figure 3 shows a comparison with the other experience selection methods. The proposed method outperformed the prior experience selection methods and achieved comparable results to the unconstrained baseline in most of the environments. In some environments, such as RoadRunner, the proposed method even accelerated the training compared to the unconstrained baseline. The proposed method showed low performance in environments where the methods that put more weights on exploration had better scores. We provide all the learning curves and the best performance score of the baseline and the proposed method for the Atari environments in Appendix F and Appendix H.

## 5.3 ABLATION STUDIES

### 5.3.1 COMPONENT ANALYSIS

We also experimented on six Atari environments with only one component of the priority metrics, i.e., surprise or on-policyness, to investigate the contribution of each factor. The results are shown in Figure 4, indicating that both components contributed to the final performance. Overall, the performance decrease was larger when the on-policyness metric was removed from the priority function. For four out of the six environments, Atlantis, BattleZone, Defender, and RoadRunner, using the on-policyness metric was enough to obtain the final results. In Enduro, removing the surprise factor slowed down the learning speed, even though using the surprise alone showed a relatively small improvement in performance.

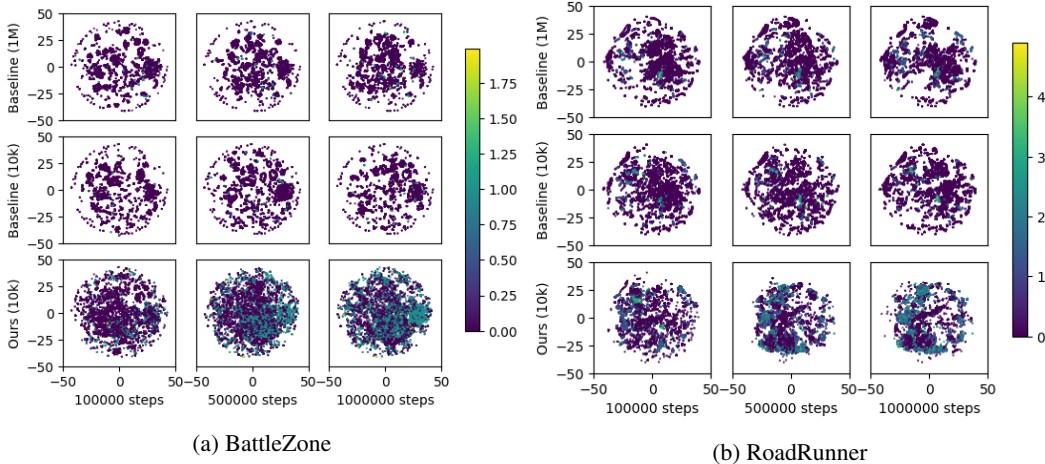

(a) BattleZone

(b) RoadRunner

Figure 5: Comparisons of latent state distributions of images stored in the replay buffer at certain training steps visualized by t-SNE(Van der Maaten & Hinton, 2008). The color of the points is determined by the cumulative discounted reward the agent received within the transition.

### 5.3.2 STATE DISTRIBUTIONS

We plotted the distributions of the latent state representations with t-SNE (Van der Maaten & Hinton, 2008), to analyze the state distributions of the experience tuples stored in the replay buffer. We also added the reward obtained during the transition in the experience tuple to the plot. We used the replay buffer from the unconstrained baseline, constrained baseline, and constrained proposed, which are the same as the ones shown in Section 5.2. The snapshots of the replay buffer were taken from every 100k training steps and all the transitions in the buffer were used for the constrained settings. As for the unconstrained settings, we used the states observed in between $(100\text{k} \times i + 90\text{k}) \sim (100\text{k} \times (i + 1))$ $(i = 0, 1, \cdots, 9)$ training steps. We converted the image state observations into latent representations by using the CNN encoder module from the unconstrained baseline model after 1M training steps. Thus, the latent representations of the observation from different experiment settings share the same latent space in the plotted results.

The results are shown in Figure 5. We can see that the state transitions that the proposed method keeps tend to have larger returns during the transitions than the baseline methods. In Figure 5a, we observed that the proposed method has a relatively higher coverage of state space compared to the baseline with a buffer size of 10k throughout the training. The state distribution of the final training steps in Figure 5b shows that the states are densely distributed in space where the baseline with a constraint has a relatively small amount of samples.

### 5.3.3 LONGER SETTING

We conducted experiments to examine whether our method would have an advantage in standard off-policy training. We examined this by using the experience selection metrics on a setting where the buffer size is set to 1M, which is a common size used for the replay buffer in off-policy training (Mnih et al., 2015; Schaul et al., 2016; Hessel et al., 2018). Since our experience selection is performed when there is no space left in the buffer, we trained the agents for 10M in each environment. We used six environments from Atari in total for this experiment. We select five environments where the proposed method performed well, and one environment where the proposed method did not show much improvement. These environments have a common property that a smaller buffer size reduces the performance of the agents.

The results are shown in Figure 6. We see a significant drop in performance for BattleZone and Qbert after around 1M steps of the training. This suggests potential negative effects of our method during a long training period. We will discuss this in detail in Section 6. The proposed method and the baseline performed almost equally well for CrazyClimber and RoadRunner. In Enduro and

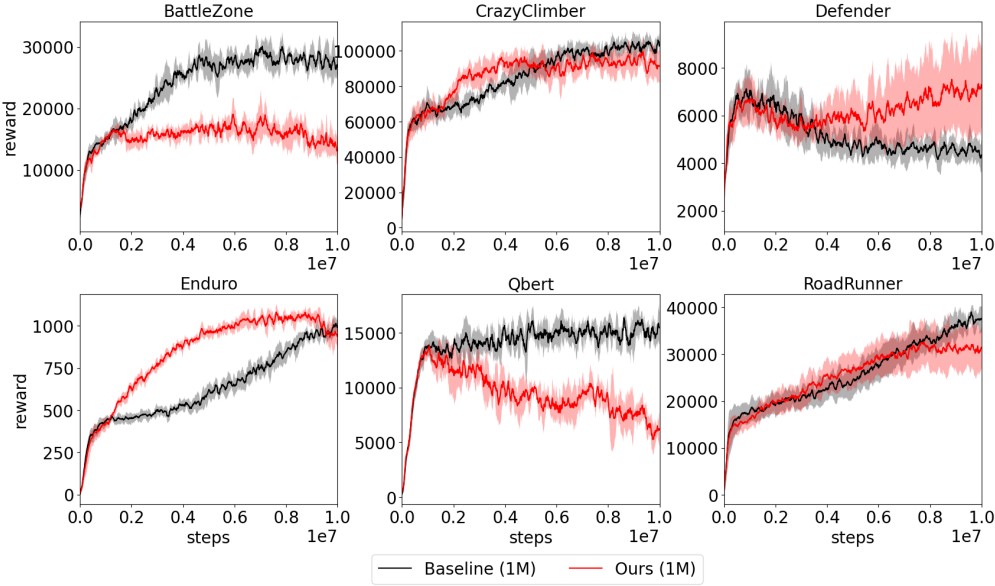

Figure 6: Comparison of the cumulative rewards with and without the proposed method when the buffer size is set to 1M. The solid line and shaded regions represent the mean and standard deviation, respectively, across five random seeds. The curves are smoothed with a moving average of 10 to improve readability.

Defender, the proposed method showed some positive effects. The former showed some acceleration in learning, while the latter showed a smaller decline in performance.

## 6 LIMITATION AND DISCUSSION

In this paper, we proposed a method to select experience tuples during training. The method successfully prevents the degradation of performance when the capacity of the replay buffer size is limited. The surprise promotes the buffer to store experience tuples that have large uncertainty and on-policyness ensures that the tuples are tied to the current policy of the agent.

One limitation of our work is the low performance of the agent in long-term training. Although using TD-errors as an indicator of learnability is reasonable to some extent, it can have negative effects when the agent is trained for a large number of time steps. For example, as the training proceeds and the overall loss decreases, the influence of the states with large loss but with small novelty will increase. For example, the initial states of episodes can have high TD-errors because the return of an episode can vary depending on the trials. Although the on-policyness metric can reduce the effect of out-of-distribution experiences, the agent can easily fall into local optima when the diversity of the data is lost. As a result, the combination of these two metrics has the vulnerability to make the state distribution converge to a small area in the state space. This makes the transition in the replay buffer highly biased. A simple example of this case is shown in Appendix B. One potential solution may be to ensure the diversity of states and actions stored in the replay buffer to some extent.

## 7 CONCLUSION

We proposed a technique to save the memory consumption of a replay buffer by using estimated importance of the experiences. Our technique is based on the idea that the importance of experiences can be estimated using the TD-errors and the closeness to the decision made by the current target policy. Our experiments demonstrated that using our method of experience retention can reduce the negative effects of a limited-size buffer. We expect that our proposed method makes RL training more applicable to low-resource environments.

## REPRODUCIBILITY STATEMENT

We provide detailed descriptions of our experimental setups and implementations in Section 5.1 and Appendix A. We also provide the source code in the supplementary materials.

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

## A   TRAINING DETAILS

The model architecture we used is based on the PFRL (Fujita et al., 2021) implementation of Rainbow algorithm. The length of each episode is 108k frames at maximum. The list of hyperparameters is shown in Table 1.

Table 1: Parameters for Atari experiments

| Parameter | Value |
|---|---|
| Batch size | 32 |
| Number of updates per training steps | 1 |
| Target update steps | 2000 |
| Replay start size | 1000 |
| Noisy net sigma | 0.5 |
| Observation size | (4, 84, 84) |
| Reward clipping | True, [-1.0, 1.0] |
| Discount rate | 0.99 |
| Action repetition | 4 |
| Learning rate | 1e-4 |
| Optimizer | Adam (Kingma & Ba, 2015) |
| Max gradient norm | 10 |
| Multi-step length | 5 |

For experiments using SEER (Chen et al., 2021), we froze the CNN encoder at 100k training steps. The initial size of the replay buffer is 10k and the capacity is increased by a factor of 2.25 after the CNN encoder is frozen and the observation states are stored as latent vectors. This is the exact gain that the replay buffer obtains when the 4x84x84 sized 8 bits int image observations are converted to 3136 length 32 bits float vectors.

## B   ANALYSIS OF FAILURE CASE IN MOUNTAIN CAR

We show some possible cases of failure using the proposed method in MountainCar-v0. In successful cases, the state action distribution of the tuples in the replay buffer will form circular curves shown in Figure 8. The curve starts from the center and ends at the top-right which refers to the bottom of the valley and the top of the hill in Figure 7.

We analyzed the state action distributions of tuples at the early and late stages of training from the failure cases using the surprise and the on-policyness metrics. At the early stage of training, surprise promoted the tuples to spread throughout the state space (Figure 9a). However, after the models are trained to some extent, the tuples start to gather at the area which is the initial position of the cart in the episodes (Figure 10a). This is because, at a certain point of training, the TD-errors from the novel states become smaller than the ones at the initial states where the variance of $Q$-values is relatively high throughout the training. We also found that the tuples selected by the on-policyness metrics can get biased, especially toward the tuples that selected the action "right" (Figure 9b). This is due to the fact that taking an action "right" on top of the hill tends to have relatively high $Q$-values compared with other actions and makes the on-policyness weight large. This also resulted in the transitions taken near the goal seen at the early stage remaining until the late stage of the training (10b). In addition, the tuples that were collected by the on-policyness metric had a possibility to form a cluster when the $Q$-value of a certain action became significantly high compared to other actions. As a result, the combination of surprise and on-policyness can result in convergence to a small area in the state space and little action diversity making the policy collapse (Figure 10c).

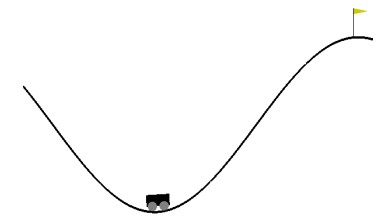

Figure 7: Appearance of MountainCar-v0

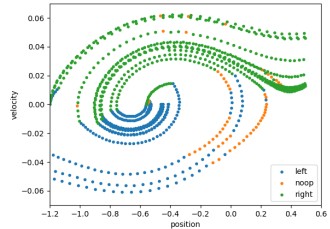

Figure 8: State action distribution of tuples in the FIFO replay buffer

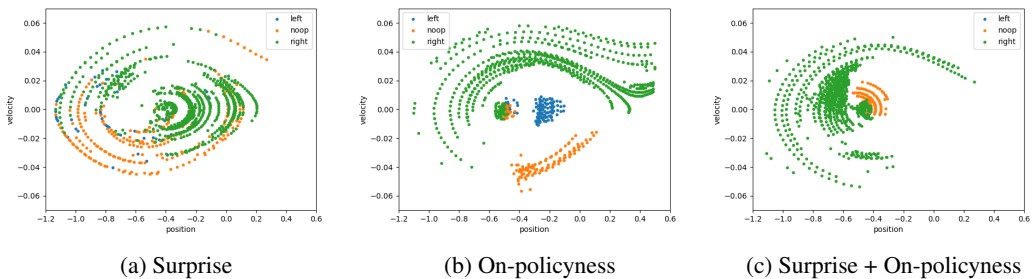

(a) Surprise        (b) On-policyness        (c) Surprise + On-policyness

Figure 9: State action distribution of tuples stored in the replay buffer at the early stage of training when using the specific pruning strategy in MountainCar-v0.

## C  APPLICATION TO DOMAINS WITH CONTINUOUS ACTION SPACE

The surprise term can be used for most off-policy algorithms. On the other hand, the on-policyness metric which needs to calculate the softmax values across possible actions at a certain state $s$ cannot be directly used in domains with a continuous action space. One possible approach would be to use the action probability that is calculated in Actor-Critic methods (Lillicrap et al., 2016; Haarnoja et al., 2018) since the on-policyness metric resembles the soft $Q$-policy.

In such a case, assuming that the action probability distribution is given as a normal distribution, the on-policyness weight can be written as:

$$w(s_t, a_t) = \mathcal{N}(a_t | \pi(a|s_t), \sigma).$$

The $\sigma$ can be either the one used in the current target policy or a fixed value. We observed that avoiding a small value for $\sigma$ when calculating the on-policyness would have slightly better performance. We set $\sigma$ to 1 throughout the training when calculating the on-policyness.

We evaluated this simple modification on Deepmind Control Suite (Tunyasuvunakool et al., 2020). We used DrQv2 (Yarats et al., 2021) as the RL algorithm and constrained the replay buffer size to 2k, which is the minimum size without changing the original exploration steps. The hyperparameters except for the replay buffer size were kept the same as in the original implementation. The results are shown in Figure 11. We saw improvements from the baseline results on most of the domains we tested on.

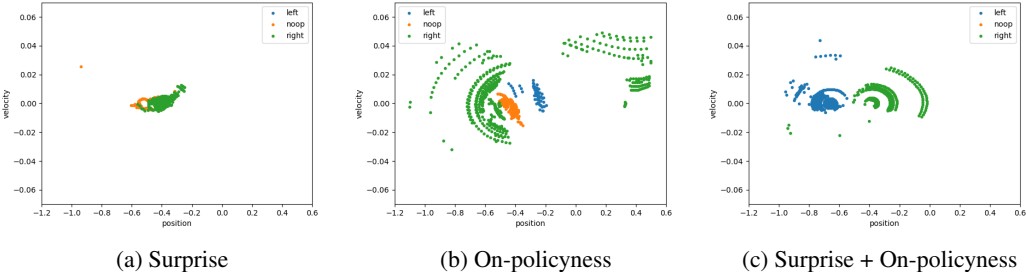

(a) Surprise       (b) On-policyness       (c) Surprise + On-policyness

Figure 10: State action distribution of tuples stored in the replay buffer at the late stage of training when using the specific pruning strategy in MountainCar-v0.

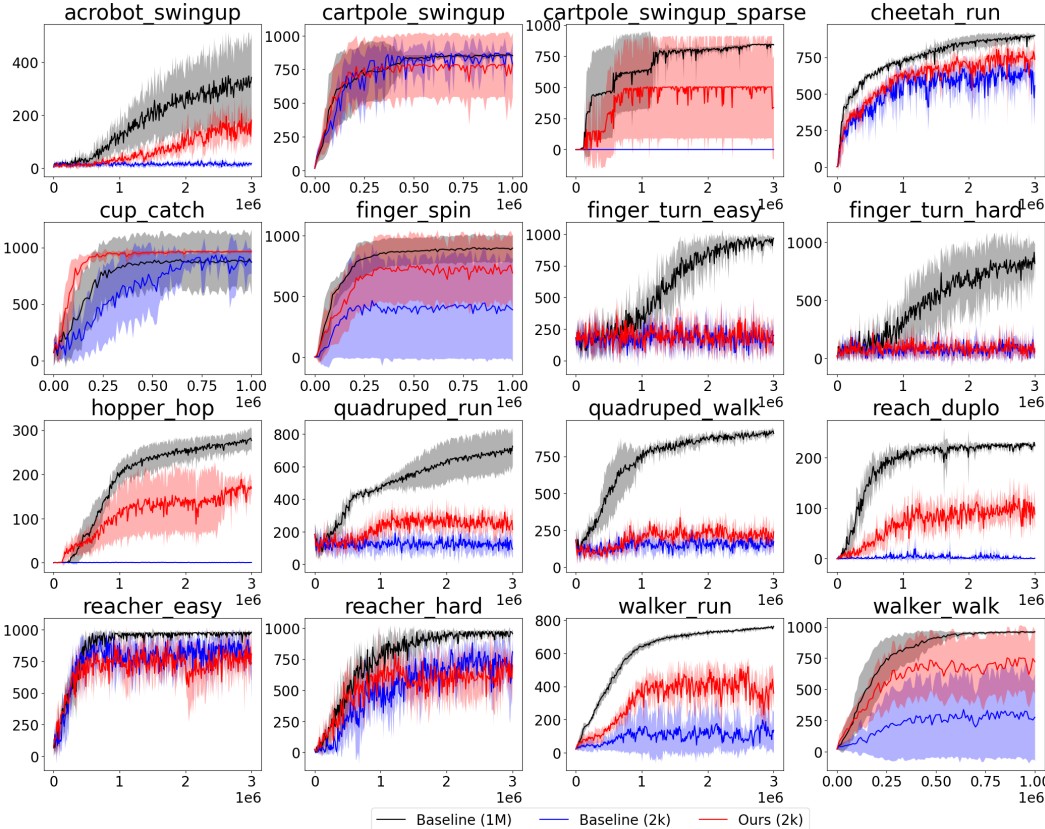

Figure 11: Comparison of the cumulative rewards with and without the proposed method when the buffer size is constrained to 2k. The solid line and shaded regions represent the mean and standard deviation, respectively, across five random seeds.

## D  DETERMINISTIC SELECTION VS STOCHASTIC SELECTION

Our proposed method deterministically selects the tuples to discard by selecting the tuple with the minimum priority value. However, this process can be modified to a stochastic procedure by using the inverse of the priority value as the sampling priority. The results of the comparisons are shown in Figure 12. Applying stochastic sampling to the proposed method caused a decrease in the performance in most of the environments. However, there were some environments where the stochastic sampling improved the performance. Those were the environments where the deterministic proposed method performed worse than the baseline. In addition, stochastic sampling had some effect on reducing the loss throughout the training.

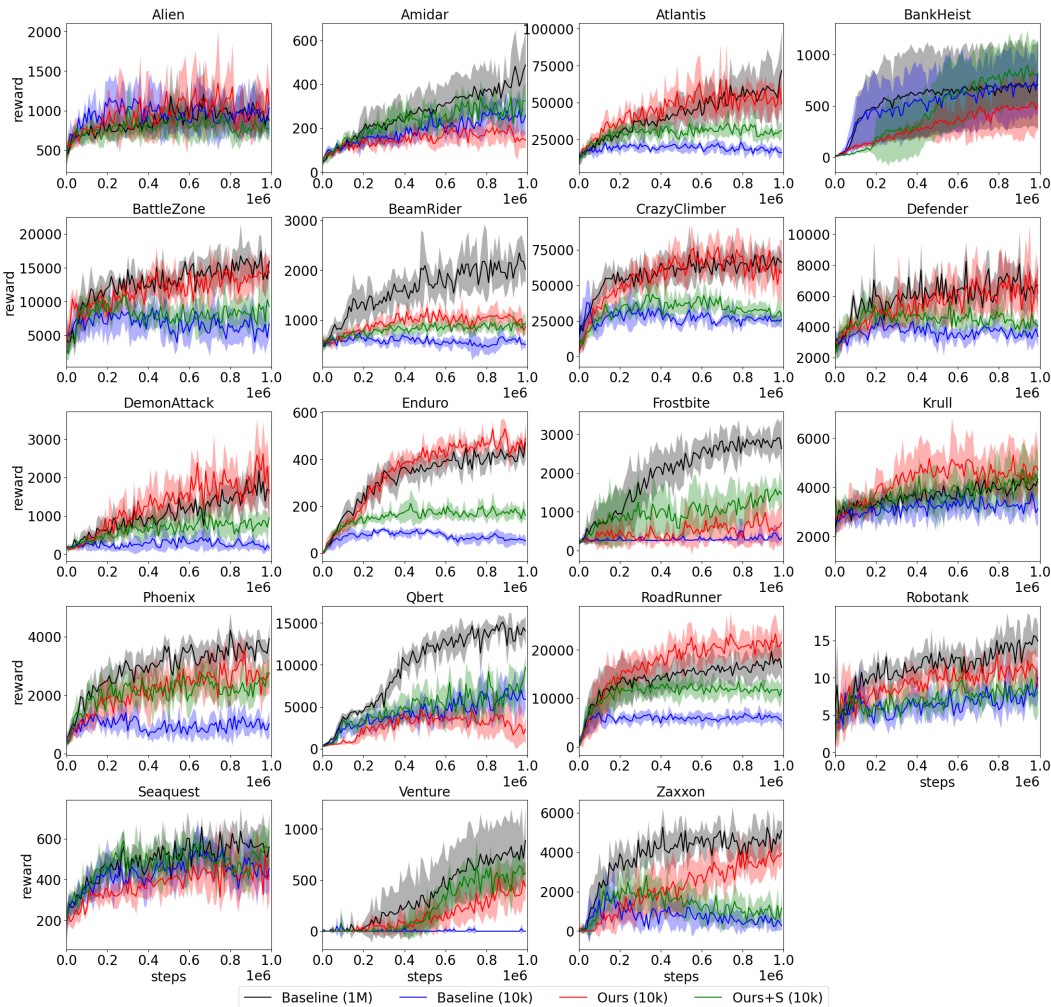

Figure 12: Comparison between the performance of deterministic and stochastic selection of the tuples. The solid line and shaded regions represent the mean and standard deviation, respectively, across five random seeds.

## E    TEMPERATURE TO SOFT-POLICY METRIC

We investigated the sensitivity of the on-policyness metric by changing the temperature parameter $\beta$. $Q$-value is multiplied by the temperature parameter $\beta$ before applying the softmax function when calculating the on-policyness. The results are shown in Figure 13. The setting with $\beta = 0$ is the same as the setting where the priority value is calculated only from surprise. We saw a slight change in the performance for most of the environments when the temperature parameter was scaled.

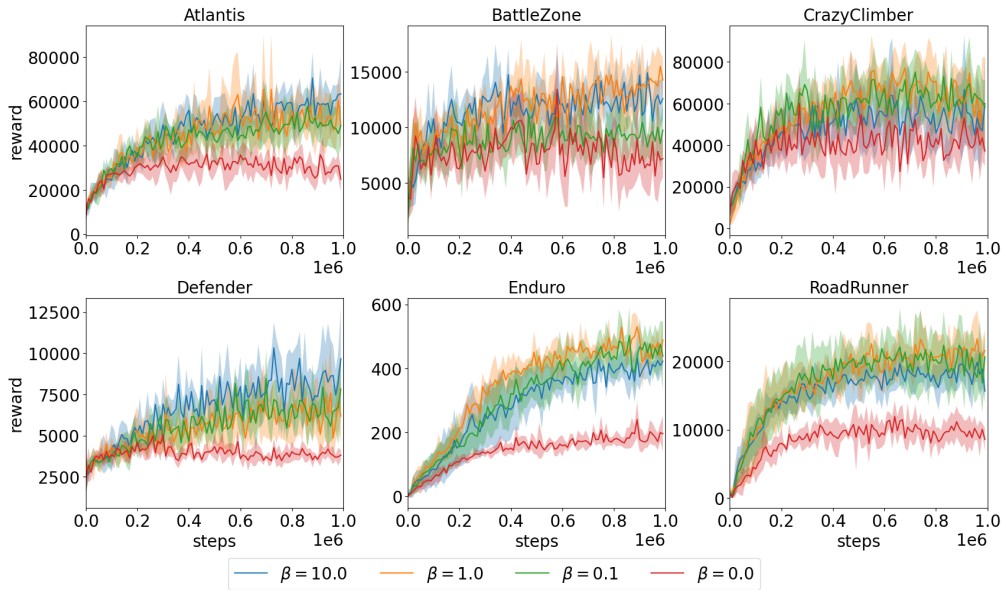

Figure 13: Cumulative rewards with various temperature values. The solid line and shaded regions represent the mean and standard deviation, respectively, across five random seeds.

## F FULL RESULTS ON ATARI ENVIRONMENTS

The learning curves of the comparison between prior methods are shown in Figure 14. The learning curves of 52 environments we experimented on are shown in Figure 15.

## G SMALLER BUFFER SETTINGS

We also experimented in Atari on settings where the size of the replay buffer is constrained to 1k, which is one-tenth the size of the original constrained setting. All the other hyperparameters except for the buffer size were kept the same as Section A. The results are shown in Figure 16.

## H BEST PERFORMANCE RESULTS

We show the mean best performance of the results shown in Apeendix F and Appendix G in Table 2.

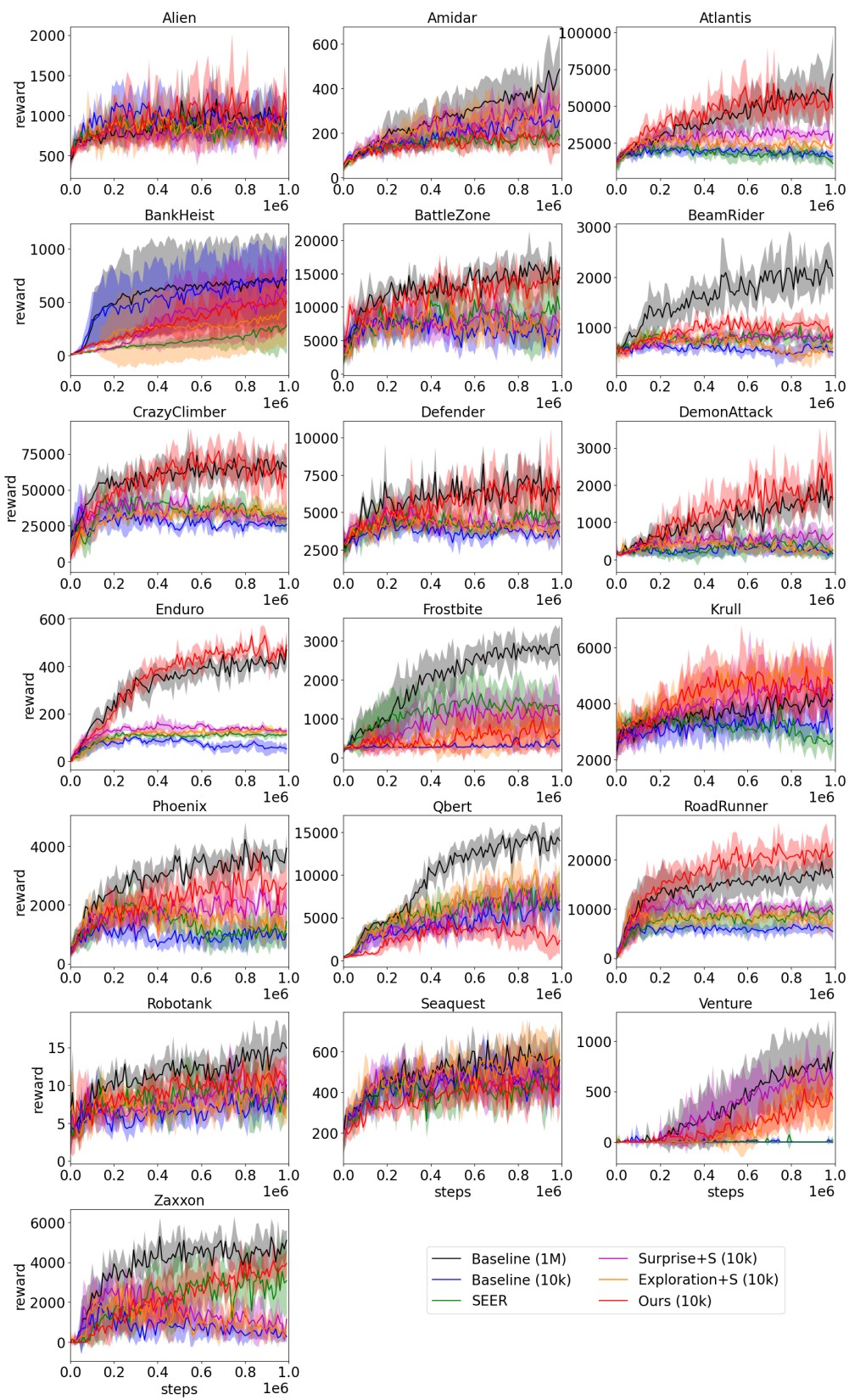

Figure 14: Comparison of the cumulative rewards in constrained-memory settings. The solid line and shaded regions represent the mean and standard deviation, respectively, across five random seeds.

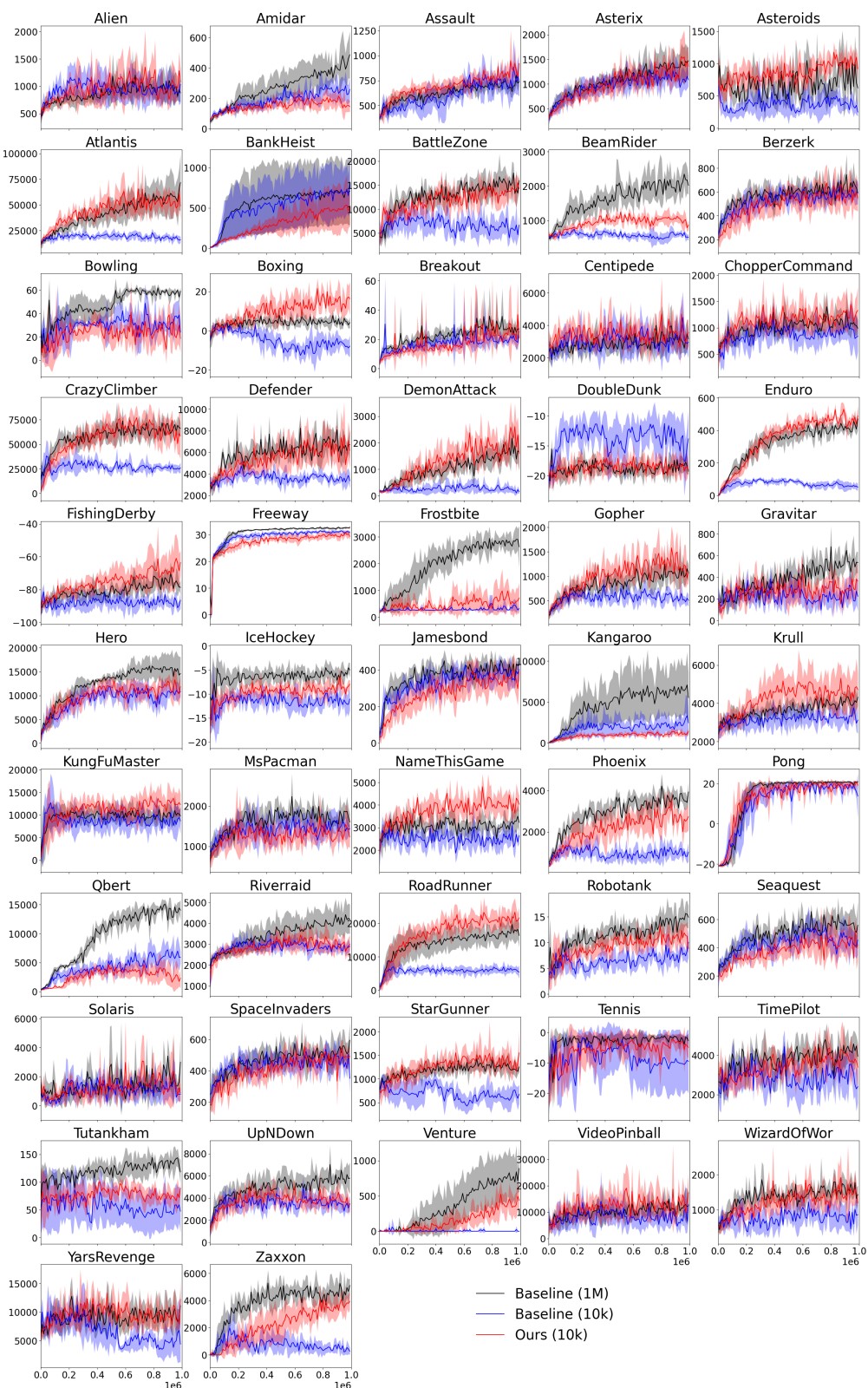

Figure 15: Comparison of the cumulative rewards with and without the proposed method when the buffer size is constrained to 10k. The solid line and shaded regions represent the mean and standard deviation, respectively, across five random seeds.

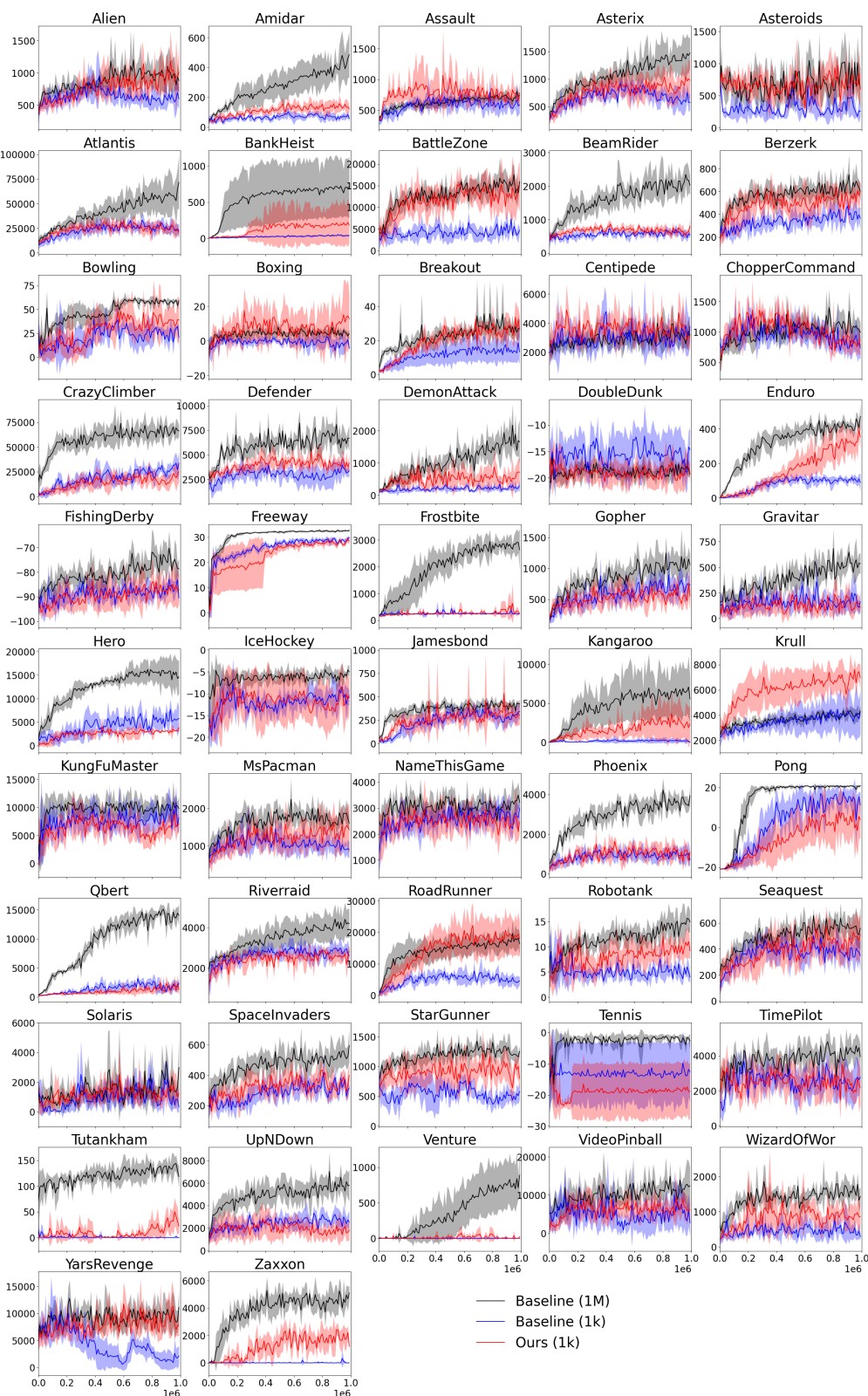

Figure 16: Comparison of the cumulative rewards with and without the proposed method when the buffer size is constrained to 1k. The solid line and shaded regions represent the mean and standard deviation, respectively, across five random seeds.

Table 2: Raw mean best scores of 52 games in Atari.The results show the mean of five random seeds, and the best results in constrained settings are indicated in bold.

| Game | Baseline (1M) | Baseline (10k) | Baseline (1k) | Ours (10k) | Ours (1k) |
|---|---|---|---|---|---|
| Alien | 1530.0 | 1475.2 | 1275.6 | **1974.4** | 1620.0 |
| Amidar | 515.2 | **357.8** | 131.0 | 276.6 | 195.8 |
| Assault | 865.6 | 1108.3 | 959.9 | 1067.5 | **1399.0** |
| Asterix | 1680.0 | 1544.0 | 1062.0 | **1890.0** | 1330.0 |
| Asteroids | 1593.2 | 1097.6 | 918.4 | **1454.0** | 1450.4 |
| Atlantis | 75500.0 | 29328.0 | 41068.0 | **79336.0** | 40276.0 |
| BankHeist | 748.4 | **895.6** | 62.0 | 604.4 | 250.0 |
| BattleZone | 20720.0 | 13800.0 | 9480.0 | 19240.0 | **23080.0** |
| BeamRider | 3110.8 | 926.2 | 900.6 | **1545.4** | 1054.7 |
| Berzerk | 843.2 | 754.8 | 574.8 | **796.0** | 760.0 |
| Bowling | 72.0 | 70.0 | 65.8 | 69.1 | **79.8** |
| Boxing | 12.6 | 6.8 | 8.3 | 28.0 | **34.0** |
| Breakout | 60.4 | **63.5** | 21.4 | 62.8 | 42.8 |
| Centipede | 5951.4 | 5985.1 | **7608.5** | 6497.5 | 6724.8 |
| ChopperCommand | 1664.0 | 1676.0 | 1660.0 | 1876.0 | **1972.0** |
| CrazyClimber | 89072.0 | 49372.0 | 43988.0 | **92172.0** | 35508.0 |
| Defender | 10026.0 | 5946.0 | 5242.0 | **9864.0** | 6500.0 |
| DemonAttack | 2669.2 | 825.4 | 762.4 | **3523.8** | 1590.4 |
| DoubleDunk | $-13.7$ | **$-8.1$** | $-8.3$ | $-14.8$ | $-10.7$ |
| Enduro | 509.0 | 124.2 | 153.7 | **600.5** | 419.0 |
| FishingDerby | $-63.3$ | $-75.2$ | $-77.0$ | **$-53.3$** | $-72.6$ |
| Freeway | 33.1 | **31.9** | 31.0 | 31.3 | 30.3 |
| Frostbite | 3350.0 | 841.6 | 764.8 | **1456.4** | 922.8 |
| Gopher | 1662.4 | 1048.0 | 1303.2 | **2024.8** | 1059.2 |
| Gravitar | 826.0 | 556.0 | 560.0 | **620.0** | 424.0 |
| Hero | 17533.2 | 13727.4 | 10981.8 | **14478.2** | 5717.2 |
| IceHockey | $-1.5$ | $-4.7$ | $-3.7$ | $-4.4$ | **$-1.8$** |
| Jamesbond | 532.0 | 498.0 | 476.0 | 472.0 | **1006.0** |
| Kangaroo | 8296.0 | **4844.0** | 632.0 | 1920.0 | 4660.0 |
| Krull | 4850.7 | 4201.4 | 5451.6 | 6125.8 | **8266.0** |
| KungFuMaster | 14136.0 | 15920.0 | 15372.0 | **18940.0** | 13312.0 |
| MsPacman | 2555.6 | 2419.6 | 1898.4 | **2564.4** | 2205.6 |
| NameThisGame | 4356.4 | 3994.8 | 3872.8 | **5288.8** | 3582.0 |
| Phoenix | 4658.8 | 2234.0 | 2004.4 | **4184.0** | 2389.6 |
| Pong | 21.0 | 20.4 | 19.8 | **20.9** | 12.4 |
| Qbert | 15897.0 | **9170.0** | 4059.0 | 6143.0 | 3854.0 |
| Riverraid | 4700.0 | **3860.0** | 3716.0 | 3832.8 | 3364.8 |
| RoadRunner | 20636.0 | 9600.0 | 11036.0 | **26224.0** | 25396.0 |
| Robotank | 19.0 | 13.5 | 15.1 | 15.6 | **16.2** |
| Seaquest | 764.0 | 676.0 | 631.2 | 628.0 | **684.0** |
| Solaris | 7211.2 | 4392.0 | 4989.6 | **5923.2** | 4156.8 |
| SpaceInvaders | 698.8 | 646.2 | 503.4 | **659.4** | 544.6 |
| StarGunner | 1764.0 | 1228.0 | 968.0 | **2124.0** | 1556.0 |
| Tennis | $-0.6$ | $-0.9$ | $-3.8$ | **$-0.8$** | $-3.5$ |
| TimePilot | 6132.0 | 5120.0 | 4936.0 | **5700.0** | 4644.0 |
| Tutankham | 159.8 | **118.0** | 29.4 | 117.3 | 64.7 |
| UpNDown | 7832.8 | 5956.8 | 4489.2 | **6098.8** | 3862.8 |
| Venture | 1008.0 | 132.0 | 20.0 | **736.0** | 128.0 |
| VideoPinball | 21752.4 | 25910.8 | 22330.6 | **29800.7** | 14075.8 |
| WizardOfWor | 2740.0 | 1768.0 | 1504.0 | **3024.0** | 2348.0 |
| YarsRevenge | 16454.1 | 16819.3 | 16543.5 | **17399.0** | 16692.9 |
| Zaxxon | 6412.0 | 3412.0 | 672.0 | **4864.0** | 3584.0 |

