# OpenReview forum: "Memory-Efficient Reinforcement Learning with Priority based on Surprise and On-policyness"
_ICLR.cc/2023/Conference — Submitted to ICLR 2023_

### Official Review · Reviewer_Lf8C · 2022-10-24

**Confidence:** 4
**Correctness:** 3
**Technical Novelty And Significance:** 2
**Empirical Novelty And Significance:** 2
**Recommendation:** 5

**Clarity, Quality, Novelty And Reproducibility:**


- One of the important points of training an off-policy algorithm is having data that is less on-policy. In order for training to progress the correlation between the data and the current policy needs to be broken to help preserve the IID assumption. How does the preference for keeping on-policy data not interfere with the off-plicy training.
- Have the authors considered instead compressing the image information? Most image information is stored as raw uncompressed data on the computer it would be good to understand the relation between this concept and that method that could instead reduce the amount of memory required by properly compressing the images in the data set.
- The method proposed in the paper is clear and concise, but it suffers from showcasing a large amount of novelty. The work could appear to be a combination of one method for computing surprise and another method for evaluating on-policiness and then just using that as a replay buffer prioritization metric. The importance of this novelty should be described better.
- Does the method proposed in the paper, in particular the measure for on-policiness, imply that it is only designed to work with the DQN algorithm and will not work for continuous action-type environments that use TD3 and SAC?
- The variance over the results in figure 3 is significant. This is implying that it's unclear if one method is performing better than the other, given the noisy results in those figures. To better understand if one method is performing better than the other, the author should run a series of statistical tests to see whether or not there is a meaningful difference between the distributions across the compared algorithms.
- Why are the Atari environments a good set of environments for the comparison of this algorithm? There is little prior work that evaluates over those particular environments and is concerned about having a lack of memory available to train a policy. It would make much more sense to try and evaluate these methods over some types of robotics or real-time environments that can have real and quite limited hardware constraints.
- Why is 10,000 experience memory samples being used for the comparisons in this paper? It is important to be able to justify this particular number and why this is a reasonable amount of experience memory compared to other values.
- It is not clear if the results displayed in figure 3 and figure 4 are converging to at least locally optimal policies. The experiments in this section should be run longer such that we can truly understand the more long-term performance comparisons between these algorithms.
- In addition, the results indicating that learning is unstable after further training is concerning. The goal of reducing the memory requirement for the algorithm is justified, but not at the cost of the training stability.


**Strength And Weaknesses:**

pros
- The method proposed in the paper is clean and very easy to understand and should be fairly easy to reproduce
- Some of the motivations are also very clear. Being able to reduce the memory requirements for the algorithms we use these days can have many important impacts on making algorithms easier to use and more widely accessible to people.

cons
- One challenge with this method also has to do with the experimental analysis that it performs on the Atari environments. There are few examples where researchers are concerned with not having enough available memory to train in the Atari environments. Including additional environments for which these issues occur would help boost the paper's motivation.
- The novelty of the method can also be seen as incremental in that it appears to be a combination of two prior well-understood methods for doing experience replay but instead is used for removing experience from the replay buffer instead of picking which ones to train the current policy.

**Summary Of The Paper:**

This paper proposes a new method for being able to train high-quality policies on Atari benchmarks while having an experienced memory that is two orders of magnitude smaller. the motivation is that storing large amounts of data and experience memory, especially for image-based environments, can be very memory intensive. If you can reduce this demand for memory, we might be able to reduce the hardware requirements needed to run these algorithms. The method proposes combining types of experience replay methods for surprise and measuring on-policiness to instead remove tuples from the replay buffer that have the lowest combination of these two metrics. The method includes some analysis on the Atari environments that may indicate that performance is better or similar to prior methods that use significantly more memory stored in their experience replay buffer.

**Summary Of The Review:**

The proposed method for reducing the memory requirement for training DQN is admerable. The analysis in the paper needs more data to understand better how the stability of the new algorithm. Also, the novelty of the method appears to be limited.

---

> ### Author Response · Authors · 2022-11-19
> **Response to Reviewer Lf8C**
>
> We thank the reviewer for their time and comments.
> > About the Atari environments
>
> Thank you. It is true that not so many researchers have to consider their memory consumption in Atari environments. However, Atari still has some complexity in the environments which makes it a challenging vision-based task and will provide some insight that is common among other robotic environments. We have added baseline results from some environments in Deepmind Control Suite in appendix.
>
> > Have the authors considered instead compressing the image information?
>
> Yes, we considered using compressing methods but did not use them because we wanted to focus on comparison between data selection methods. For example, SEER discussed in related work stores the image observation as latent vectors obtained from the CNN encoder. Since there was a few suggestions to add it as comparison, we added results of SEER.
>
> > Does the method proposed in the paper, in particular the measure for on-policiness, imply that it is only designed to work with the DQN algorithm and will not work for continuous action-type environments that use TD3 and SAC?
>
> Since the on-policy metrics resembles the soft $Q$-policy, one approach to apply our method to continuous settings would be directly using the action probability from the policy function. We have added some baseline results in Deepmind Control Suite using DrQv2 which runs on the DDPG algorithm.
>
> > Why is 10,000 experience memory samples being used for the comparisons in this paper?
>
> We used 10k as the size of the replay buffer because the buffer size of around 10k had a performance gap compared with the performance of the unconstrained setting. In addition, 10k is briefly the number of tuples where the memory consumption would go below 1GB.
>
> >  The experiments in this section should be run longer such that we can truly understand the more long-term performance comparisons between these algorithms.
>
> That is true. We followed previous work, including SEER, that evaluates methods on 500k steps.

---

> > ### Comment · Reviewer_Lf8C · 2022-11-20
> > **Updates appreciated**
> >
> >
> > Re: results
> >
> > The experiments on Atari are helpful to indicate the method can help, but it is important to show the method is not biased toward Atari-type environments. For example, the updated results appear to show that the method proposed in the paper does not perform as well in deep mind control suite tasks. Did these DM control suite experiments use image-based inputs?
> >
> > Re: comparisons to prior work
> >
> > The addition of SEER is appreciated. However, it is important to discuss more why SEER is better for comparison and discuss more about the memory efficiency of each method compared to performance.
> >
> > Re: Why is 10,000 experience memory a good size?
> >
> > The reasoning for this choice still sounds unjustified. The motivation for the work is to compress data for applications that may have limited memory. One would think the reason for choosing 10k should be motivated by some connection to the hardware limitation of embedded computers or robotics.

---

### Official Review · Reviewer_FE4B · 2022-10-27

**Confidence:** 4
**Clarity, Quality, Novelty And Reproducibility:** Good.
**Correctness:** 3
**Technical Novelty And Significance:** 3
**Empirical Novelty And Significance:** 2
**Recommendation:** 5

**Strength And Weaknesses:**

The paper is well-motivated and the writing is really good. The proposed method is indeed much more memory-efficient than Rainbow in Atari games when agents are trained with 1M steps.

However, as also pointed out by the authors, when agents are trained longer (10M steps), the proposed algorithm actually performs much worse than the baseline in some tasks, even using a replay buffer with the same size as the one in baseline. The discussion of the limitation in Section 6 is not convincing or not supported by experiments. This paper can be much better if further experiments are conducted to show and analyze the limitation of instability with longer training. Moreover, a study of the affect of different buffer sizes (e.g., 1k, 10k, 100k, 1M) is necessary in order to see the whole picture.

Minor issues and questions:
- In Section 4.3, the function $f^{priority}$ is not defined.
- In Figure 5(a), it seems that the state coverage of the proposed method (with 10K samples) is much better than the baseline (with 1M samples). How is it possible? Or maybe I interpret it incorrectly.
- In Algorithm 1, will all priority scores in the replay buffer be recomputed after the Q function is updated? If so, will this slow down the training process? Can you show the wall-clock training time? If not, since the priority calculation function depends on the current Q value function, the pseudo-code hints that all priority scores in the replay buffer are recomputed, as shown in Line 8 & 11. It might be better to replace $f(\tau)$ with some other symbol to show that the scores are not recomputed in Line 8 & 11.

Suggestions
- Instead of moving straight to Atari games, conducting experiments in toy environments might provide more insights. For example, no results are presented to support the claim that "The surprise promotes the buffer to store experience tuples that cover the state space" in Section 6. This claim can be better verified in low-dimensional tasks, such as Mountain Car.
- My personal experience suggests that a less frequent update for Q function helps improve agent performance when the replay buffer is small.
- In Figure 4, it seems that adding "surprise" is not that helpful and "on-policyness" is main reason for performance improvement. Did you try removing "surprise" (thus reduce negative effects as mentioned in Section 6) and redo the experiments in Figure 6?
- Some methods listed in this [paper](http://arxiv.org/abs/1809.05922) might help to improve the performance of the proposed algorithm, such as reservoir sampling.

**Summary Of The Paper:**

This paper aims to make off-policy RL algorithms more memory-efficient by reducing the size of the experience replay buffer. Specifically, the proposed algorithm keeps experience tuples with higher importance and discards unimportant ones. The importance of a tuple is measured by its "surprise" (TD error) and "on-policyness". Experimental results in Atari games show that the proposed algorithm achieves comparable performance with only 1% size of the original replay buffer, compared with the baseline method (Rainbow).

**Summary Of The Review:**

Overall, this paper is well-motivated, focusing on an important problem. However, the performance drop of longer training limits the application of the proposed method. I believe this paper can be further improved with more analytical experiments in toy tasks.

---

> ### Author Response · Authors · 2022-11-19
> **Response to Reviewer FE4B**
>
> Thank you for your time and helpful suggestion on our paper.
> > The function $f^{priority}$ is not defined.
>
> Thanks for pointing it out. We will correct it to $f(\tau)$.
>
> > State coverage in Figure 5(a)
>
> The state distribution shown in the figure is the ones that were observed in the recent 10k steps. Thus the whole state distribution of the replay buffer in unconstrained methods will be the union of the results from the previous time steps, so the state distribution will be much larger than the plotted results.
>
> > Algorithm 1
>
> Only the tuples that were sampled as a batch is updated. Thank you for your suggestion. We have changed the notation in the pseudo-code so that the tuples with the minimum value are searched for based on the computed value. In our Atari experiments, the overhead was around 20% increase in the training time.
>
> > Suggestions
>
> Thank you for your valuable suggestions. We have provided experimental results on Mountain Car to show some insight on why the proposed method performed worse than the baseline.

---

> > ### Comment · Reviewer_FE4B · 2022-11-24
> > **Thank for your updates**
> >
> > Thank you for providing analysis on Mountain Car. Given these results, it seems that both metrics (on-policyness and surprise) are affected by the magnitude of Q values. Is it possible to mitigate the negative affect of Q values by dividing them?

---

> > > ### Author Response · Authors · 2022-11-28
> > > **Response to Reviewer FE4B**
> > >
> > > Thank you for your question.
> > >
> > > > Is it possible to mitigate the negative affect of Q values by dividing them?
> > >
> > > It might be possible to reduce the negative effects by dividing them but it would be difficult since the $Q$-values can take negative values.
> > > We have tried dividing the Q-values by changing the numerator of the on-policyness metric as $\exp (\frac{Q(s_t, a_t) - \min_a Q(s_t, a)}{\max_a (Q(s_t, a_t) - \min_a Q(s_t, a))} )$, but it did not improve the overall performance. As for the surprise, it is difficult to divide the values since they are calculated using the KL-divergence between the distributions of $Q$-values.

---

### Official Review · Reviewer_HQEE · 2022-10-28

**Confidence:** 4
**Correctness:** 3
**Technical Novelty And Significance:** 2
**Empirical Novelty And Significance:** Not applicable
**Recommendation:** 5

**Clarity, Quality, Novelty And Reproducibility:**

The clarity and quality of experiments in this paper are very good. It was easy to read, and everything was well motivated. Reproducibility seems good, as they provide the code for their experiments. In terms of novelty, this work does seem a bit incremental, but still novel.

**Strength And Weaknesses:**

Strengths:
- This paper is very well written, easy to follow, and well motivated.
- The experiments are done on a wide range of the Atari games. For most of the games (although not all), this method does outperform the simple FIFO replacement strategy of a comparably sized buffer, and is able to close the performance gap with a larger sized buffer.
- The ablation experiments show the contributions of each component in the priority weight, and show that both surprise and on-policyness do contribute to better performance.
- This paper also tried running on longer timescales, with buffer size the same as what is typically used for Atari in the literature, and showed that their method actually does worse than traditional buffers on several environments on this longer timescale. While this is a limitation, it's good that the authors do talk about it, and it potentially highlights a future direction of work.

Limitations/Questions:
- How does your method work with n-step returns in Rainbow? If you remove a transition in the middle of a trajectory, does that invalidate several other transitions as well? If so, it might be worth exploring this relationship of the number of invalidated transitions and the horizon value in n-step, and see if a replacement strategy could be devised to work around that.
- As mentioned above, the paper does mention limitations of the method when learning in longer timescales.

Edit:
After the discussion with other reviewers, I am lowering my score to marginal reject. I still think that this is an idea worth pursuing, but I think you need a few more experiments/analysis.
- As other reviewers pointed out, using 10k is a bit arbitrary. You should do a sweep over different replay sizes to figure out the effect of replay size on different tasks.
- If you are using Atari, you should show some runs with the full 50m steps.


**Summary Of The Paper:**

This paper introduces a mechanism for selecting transitions to remove/replace from the replay buffer during training of an RL agent. The samples are selected based on a priority function that is a multiplication of surprise (TD-error) and a on-policy weight of the transition. Using this priority function, the paper claims that for shorter settings, their approach outperforms agents with comparably sized replays and is able to match agents with much larger replays on most Atari games.

**Summary Of The Review:**

This is a well written paper with good experiments supporting their methods ability enable learning of agents with small replay buffers at short timescales. I recommend acceptance.

---

> ### Author Response · Authors · 2022-11-19
> **Response to Reviewer HQEE**
>
> Thank you for your time and feedback on our paper.
> > How does your method work with n-step returns in Rainbow?
>
> The states contained in the tuples are ($s_t$, $s_{t+n}$), so the removing a transition would not invalidate other transitions.

---

### Official Review · Reviewer_f3A3 · 2022-10-30

**Confidence:** 5
**Correctness:** 3
**Technical Novelty And Significance:** 2
**Empirical Novelty And Significance:** 2
**Recommendation:** 3

**Clarity, Quality, Novelty And Reproducibility:**

The priority metrics proposed in this paper are mainly based on previous works (i.e. TD-error in "Prioritized experience replay" paper, policyness weight in "Soft-actor critic"), but their combination can be considered new. However, since this is an empirical paper, the results should provide a strong signal about the significance of this method which they don't at the moment.

**Strength And Weaknesses:**

While the idea of only saving high valued samples in the reply buffer and using smaller replay size is interesting and can be valuable, I have a couple of concerns with this paper:

- Computational complexity:
The main motivation for this paper is to "make RL training more applicable to low-resource environments''. Although this is the right problem to study,  the current method doesn't come for free and it increases computational complexity of a method as priority function requires to search through the replay buffer at each time step and estimate the score ( search can be more expensive here). The paper  doesn't analyze the complexity of the proposed method and does't provide insights about computation and memory trade-off.

- Baseline methods:
In section 5.1, even though authors discuss various baseline methods, they compare their method with these baseline only on 6 environments. While I understand running Atari is expensive, since the contribution of this paper heavily relies on the results, the current results are not conclusive. I'd like to note that authors reported results with smaller replay buffers but the same rainbow method, these results are not as important as comparing with other baselines. Moreover, important baseline like "Improving computational efficiency in visual reinforcement learning via stored embeddings." is missing in the experiments. Also, likelihood free ration and importance sampling weighting can be used as other baselines.


- Figure 2:
It is not clear to me what the baseline method in Figure 2? If it is 1K or 10K baseline, they are not appropriate baseline methods. It should be compared with methods which are discussed in the paper.

- Validity of Results:
Comparing Baseline (1M) reported in Figure 8 with Dopamine ( https://google.github.io/dopamine/baselines/atari/plots.html), I notice a huge discrepancy between this paper's results and Dopamine which makes me anxious about validity of this results. Take Gopher as example, Dopmaine shows rainbow gets to average return of 10000 after 200M samples while this paper shows 1000. The same is true about other environments.

- Writing:
While the paper is written well in general, there are notable issues. Q-function is distribution but the equations treat it as not being a distribution. I think this should be fixed. Also, this paper discusses continual learning in the related work sections which I have a hard time to find any relationship with. Instead, this paper should have focused on related works which are relevant to this paper.

**Summary Of The Paper:**

This paper proposes a method to make Rainbow more memory efficient. In particular, it proposes a way to only save data which seems more important using the combination of on-policyness and surprise metrics. Surprise metric is built based on TD-error and on-policyness measures how far behavioral policy is from target policy. One major difference between priority replay buffer and this method is priority replay buffer does not discard samples and saves all data until capacity allows; however, this method discards samples which are unimportant. To show the effectiveness of their method, this paper uses 52 games of Atari benchmark.

**Summary Of The Review:**

The main and critical concern about this paper is the experiments which are not conclusive considering the fact that this is an empirical paper. Another concern is this paper only studies pixel-based environments.  I don't understand why this paper can not be used with state-based environments like MuJoCo. I'd make my final recommendation after rebuttal as I'd like to hear the authors' rebuttal. That being said, this paper does not meet ICLR acceptance bar as it stands.

---

> ### Author Response · Authors · 2022-11-19
> **Response to Reviewer f3A3**
>
> Thank you for reviewing our paper and providing detailed feedback.
> > Computational Complexity:
>
> In our Atari experiments with batchsize=32 and replay\_buffer\_size=10k, the overhead was around 20\% increase in the training time.
> Since the priority value is calculated when the pruning is performed, the computational complexity will be determined by the data structure of how the priority values are stored and updated. For example, one can implement it as an array-like structure. This will take $O(N)$ for searching and $O(B)$ for updating the priority value, where $N$ is the length of the replay buffer and $B$ is the batch size. In our experiments on Atari, we used a min-tree structure that consumes O(1) for determining the min value, $O(B\log{N})$ for finding the tuple which has the minimum value and $O(B\log{N})$ for updating the value. The user can select whatever data types depending on the replay buffer size and the batch size they use to train the agent.
>
> > Baseline methods:
>
> We have added the results from SEER.
> One reason we did not use SEER for comparison was that we wanted to focus on data selection methods. In addition, SEER needs to store latent vectors and requires a CNN encoder with small-length vectors, which  makes it difficult to perform a fair comparison. In the original paper, SEER used a small encoder which outputs a latent vector consisting of 576 scalar values. In our case, the latent representation consists of 3136 scalar values. The raw observation is 4 x 84 x 84 x 1 bytes = 28224 bytes. A latent vector is 3136 * 4 bytes = 12544 bytes.
>
> > Figure 2:
>
> The results in figure 2 are showing the baseline results from 10k.
>
> > Validity of Results:
>
> It is true that the scores that the agent obtained in 1M steps training is lower than that of the 50M steps training. However, we followed previous work, including SEER, that evaluates methods on 500k steps.
>
> > Writing:
>
> Thank you. It might have been confusing to mention that the Q-values are treated as disctributions in section 3.1. We treat $p(Q)$ as the distribution of $Q$-value and $Q$ as the scalar of the $Q$-value computed by taking the expectation.
>
> > Experiments:
>
> Our method does not limit the domain to pixel-based domains. However, the main domains where the memory consumption is considered would be in pixel-based domains. For the conclusiveness concern, we have added extra experiments on Atari in environments where the SEER was evaluated and there was a decrease in performance when the buffer size was reduced. We also added some baseline experiments on DMC.

---

### Decision · Program_Chairs · 2023-01-20

**Decision:**

Reject

**Justification For Why Not Higher Score:**

The paper has many deficiencies as described above.

**Justification For Why Not Lower Score:**

N/A

**Metareview: Summary, Strengths And Weaknesses:**

This paper addresses the problem of maintaining an expensive large replay buffer in applying deep reinforcement learning with limited resources. To address this problem, the authors propose keeping a smaller replay buffer where samples with two specific attributes are kept: surprise and on-policy-ness. A specific method for measuring these two attributes for a transition is provided, and experiments are given to support the method.

Several reviewers agree that the paper is well-written and motivated. However, the reviewers showed several concerns. I met all the reviewers to discuss the paper. Most of us agreed that the paper addresses an important issue. Among many great things about deep RL, its excessive resource requirements are not one of them. It is desirable to see if the same performance of the existing deep RL methods can be achieved using less amount of resources. Doing so frees up the resources for other uses for the haves and makes previously unavailable tools available for the have-nots.

However, whether the proposed method contributes significantly to this goal remained a question. Reviewers wanted a decisive message and a clear value from the experimental results.

We have specific recommendations for improving the significance of the contribution.
- Experiments for Atari are shown with only 1M steps, whereas common benchmarking on these tasks is done with much longer runs. Hence, we suggest running the experiments with longer runs. To keep the size of the experiments manageable, it would be better to use less number of tasks instead. For Atari environments, this is the most commonly used setup. Hence, a successful resource reduction should be shown in that case.
- The choice of buffer size 10K should be well justified. A good step would be to perform a sweep of different buffer sizes. This may reveal a comparative and consistent trend between the buffer size and performance of DQN and the proposed method.
- When achieving a small buffer size is the goal, PPO becomes relevant. Hence, quantitative or qualitative comparison and discussion of PPO are important here.
- It is worrisome that the proposed method does not work well in other conditions, such as with bigger buffers, longer runs, or DM control suite tasks. Should we really be excited about this proposed method, then? These issues should be addressed and discussed.

We strongly recommend you consider these improvements and resubmit to the next venue.



**Summary Of Ac-Reviewer Meeting:**

The reviewers generally agreed that the paper has many issues, stopping the work from giving a decisive message and a clear value.